# Is Histamine and Not Acetylcholine the Missing Link between ADHD and Allergies? Speer Allergic Tension Fatigue Syndrome Re-Visited

**DOI:** 10.3390/jcm12165350

**Published:** 2023-08-17

**Authors:** Hilario Blasco-Fontecilla

**Affiliations:** 1Department of Psychiatry, School of Medicine, Autonoma University of Madrid, 28049 Madrid, Spain; hmblasco@yahoo.es; Tel.: +34-911916012; 2Department of Psychiatry, Puerta de Hierro University Hospital, Health Research Institute Puerta de Hierro-Segovia de Arana (IDIPHISA), Majadahonda, 28222 Madrid, Spain; 3ITA Mental Health, Korian, 28043 Madrid, Spain; 4Center of Biomedical Network Research on Mental Health (CIBERSAM), 28029 Madrid, Spain

**Keywords:** diamine oxidase (DAO) deficiency, attention deficit hyperactivity disorder (ADHD), histamine, cognition, attention

## Abstract

Speer allergic tension-fatigue syndrome (SATFS) is a classic allergy syndrome characterized by allergy-like symptoms, muscle tension, headaches, chronic fatigue, and other particular behaviors that were initially described in the fifties. The particular behaviors displayed include symptoms such as hyperkinesis, hyperesthesia (i.e., insomnia), restlessness, and distractibility, among others. Interestingly, these symptoms are very similar to descriptions of attention deficit hyperactivity disorder (ADHD), the most prevalent neurodevelopmental disorder worldwide, which is characterized by inattention, hyperactivity, and impulsivity. The clinical description of SATFS precedes the nomination of ADHD in 1960 by Stella Chess. In this conceptual paper, we stress that there is a gap in the research on the relationship between ADHD and allergic pathologies. The hypotheses of this conceptual paper are (1) SATFS is probably one of the first and best historical descriptions of ADHD alongside a common comorbidity (allergy) displayed by these patients; (2) SATFS (ADHD) is a systemic disease that includes both somatic and behavioral manifestations that may influence each other in a bidirectional manner; (3) The role of neuroinflammation and histamine is key for understanding the pathophysiology of ADHD and its frequent somatic comorbidities; (4) The deficiency of the diamine oxidase (DAO) enzyme, which metabolizes histamine extracellularly, may play a role in the pathophysiology of ADHD. Decreased DAO activity may lead to an accumulation of histamine, which could contribute to core ADHD symptoms and comorbid disorders. Further empirical studies are needed to confirm our hypotheses.

## 1. Introduction


*“One of the most significant developments in contemporary medicine has been the increasing recognition that the soma and the psyche are not distinct and unrelated, but are so closely bound together as to make proper investigation of both a necessary part of the management of disease.”*
Frederic Speer (The allergic tension-fatigue syndrome, 1954)

In 1954 and 1958, Frederic Speer, head of Pediatric Allergies at the University of Kansas, published two seminal studies focused on a new clinical entity, allergic tension-fatigue syndrome [1,2]. Speer allergic tension-fatigue syndrome (SATFS) can be defined as a syndrome characterized by (1) physical symptoms, including a constitutional allergic state, hyperhidrosis, pallor, infraorbital edema, headaches, and excessive lacrimation, salivation, and bladder tone [1,2]. The allergy-like symptoms, such as rhinitis and skin rashes, could be either triggered or worsened by exposure to certain foods or environmental factors; and (2) behavioral symptoms, including tension (i.e., restlessness, overactivation, hyperkinesis, aggressive irritability, apprehensive, antisocial), fatigue (i.e., lethargy, sluggishness, tiredness, boredoms interfering child’s progress in learning, oversensitiveness), and achiness (i.e., aching of the leg, arms, neck, and back). Speer stated that “Allergic tension may be defined as a clinical allergic state that is marked by diffuse neuropsychic overactivity. It includes both a motor component (hyperkinesis) and a sensory component (hyperesthesia). Usually, both hyperkinesis and hyperesthesia are present in the oversensitive allergic child” [2]. Interestingly, the particular behaviors displayed by either children or adults, including symptoms such as hyperkinesis, hyperesthesia (i.e., insomnia), restlessness, and distractibility, are very similar to recent descriptions of attention deficit hyperactivity disorder (ADHD), the most prevalent neurodevelopmental disorder worldwide [3]. ADHD symptoms typically encompass hyperactivity, inattention, and impulsivity. At the time that Speer published his work, the concept of ADHD was still emerging. As a matter of fact, Stella Chess was one of the first authors to nominate this syndrome in 1960 [4], and the concept that alludes to what we know today as ADHD only appeared in the third version of the Diagnostic and Statistical Manual of Mental Disorders (DSM-III).

Speer was a pioneer of the psychosomatic approach. In 1954, he suggested a disruptive view of the psyche–soma relationship. At this time, the psychosomatic approach defended by alienists appeared to operate in just a single direction (the psyche affecting the soma, i.e., headaches are secondary to depression). On the contrary, Speer stressed that “this relationship may operate in the opposite direction, and that somatic disease may often be the cause of disorders of feeling and behavior rather than the effect” [1]. He reviewed the scientific literature available at this time and provided compelling evidence about the possibility that “nervous system disturbances” (behavior) could “constitute a primary allergic syndrome” [1]. In his second paper, published in 1958, Speer again stressed this idea (“the behavior disturbance of allergic children constitutes a primary allergic disorder of the nervous system”) [2]. In 1970, in an even less-known paper, he linked the clinical manifestations of SATFS with the cholinergic system [5]. Unfortunately, the work by Speer was not given particular interest until the next ground-breaking step in this field.

In 1989, Paul Marshall recognized the absence of systematic studies testing the effects of allergies on ADHD behaviors [6]. However, he also stressed that the limited evidence was convergent in finding a particular relationship between allergies and ADHD in some hyperactive patients. Furthermore, he asked himself, “if these behaviors are caused directly by allergic reactions, what are the biochemical or brain mechanisms involved in the pathogenesis of ADHD with an etiology of allergy?” [6]. After a thorough review of the literature, he postulated a biochemical model (cholinergic–adrenergic activity imbalance), explaining the relationship between allergies and ADHD (“when the cholinergic/adrenergic activity balance is tilted in the direction of significant cholinergic predominance, the following behaviors typically occur: fatigue, sleepiness, psychomotor retardation, dysphoria, slowed thinking, and problems in maintaining alertness and attention […]. With respect to behavioral arousal, excessive cholinergic activity could lead to hypoactivity and, in the extreme, psychomotor retardation and fatigue […])” [6].

Unfortunately, none of these authors stressed the role of histamine as a potential mediator in the relationship between ADHD and the immune system (including inflammation) in general and allergies in particular. This is, at least, surprising, as histamine has played a core role in allergies since the use of antihistamine drugs since the beginning of the 20th century.

The latest step in the liaison between allergies and ADHD took place in 2009 when Pelsser et al. [7] postulated the hypothesis that ADHD was a (non) allergic hypersensitivity disorder. However, even these authors restricted the role of histamine to the effect of food additives on behavior “via a non-IgE-dependent histamine release from mast cells and basophilic granulocytes”.

To sum up, there is a research gap that needs to address all these caveats regarding the relationship between allergies and ADHD: what happened with the concept of SATFS?; might the initial descriptions of SATFS be just a description of an emerging syndrome in the sixties, namely, ADHD?; if so, could histamine be the missing link between the classic ADHD symptomatology and the frequent somatic comorbidity reported in patients diagnosed with ADHD?

The hypotheses of this conceptual paper are

**Hypothesis** **1.**
*The clinical descriptions made by Speer in 1954 and 1958 about an “allergic tension-fatigue syndrome” is probably one of the first and best historical descriptions of patients with ADHD affected by a comorbidity (allergy) frequently displayed by these patients;*


**Hypothesis** **2.**
*SATFS (ADHD) is indeed a systemic disease including both somatic and behavioral manifestations that may influence each other in a bidirectional manner (i.e., psyche influences soma, and soma influences psyche: allergy and allergic treatments such as antihistamine drugs influence ADHD behavioral symptoms, and reversely, core ADHD symptoms and ADHD treatments influence allergic symptoms);*


**Hypothesis** **3.**
*The role of histamine is key for understanding the pathophysiology of the typical medical comorbidities displayed by patients with ADHD and complements the cholinergic–adrenergic disbalance theory of the ADHD–allergy relationship; and finally,*


**Hypothesis** **4.**
*The diamine oxidase (DAO) enzyme, which metabolizes histamine extracellularly, may play a key role in the pathophysiology of ADHD. A decreased DAO activity may lead to an accumulation of histamine, which could contribute to ADHD symptoms.*


We begin by describing the clinical picture (behavioral and somatic symptoms) of SATFS (Section 2.1 and Section 2.2). Next, I describe the underlying cholinergic/adrenergic activity disbalance postulated to explain the relationship between allergies and ADHD (Section 2.3). Later on, I summarize the role of histamine in the body and brain (Section 3). Finally, I re-visit the SAFTS as a typical clinical picture of ADHD and present emerging evidence of the key role that histamine and the DAO enzyme may play in both ADHD and the typical somatic comorbidities presented by patients with ADHD; in other words, I include literature backing up the four hypotheses (Section 4).

## 2. Speer Allergic Tension-Fatigue Syndrome (SATFS)

As said before, SATFS includes behavioral and systemic constitutional manifestations.

### 2.1. Behavioral Allergic Traits

In his seminal works (1954, 1958), Speer reported that some allergic patients, either children or adults, tended to initially “overdo and overreact”, and then to become strongly fatigued [1,2]. In other words, he suggested that a distinct pattern of behavior was typical of some allergic individuals. He included this pattern of behavior within SATFS [1,2]. Both behavioral components of SATFS (tension and fatigue) had two aspects: (1) a motor component; and (2) a sensory component [1,2]. The symptoms of both components are summarized in Table 1. As we will defend later (see Section 4, first hypothesis), most behavioral symptoms of SATFS are typical symptoms of ADHD, the most prevalent neurodevelopmental disorder worldwide [3]. Unfortunately, Speer never had the opportunity to suggest this, as the concept of hyperactivity was initially nominated by Stella Chess in 1960 [4], and ADHD as a syndrome only appeared in the third version of the Diagnostic and Statistical Manual of Mental Disorders (DSM-III) in 1980.

In his original description of the behavior of people with SATFS, Speer said, “While some victims of allergic fatigue may be said to be tired, others may more properly be regarded as torpid. Rather than muscular fatigue, there seems to be a depression of sensory and psychic function, and affected children may be described as sluggish, inactive, sleepy, and apathetic” [2].

### 2.2. Constitutional Allergic Traits

In addition to the disturbed (tension-fatigue) behavior typical of allergic children, Speer also considered the “constitutional allergic traits”. In other words, the original constitutional, systemic syndrome depicted by Speer included not only allergic and motor fatigue but also torpor, “asthma, gastrointestinal allergy, eczema, and nasal allergy” and a “less well defined and less well known phenomena” that he interpreted as “evidence of increased parasympathetic or cholinergic activity” which included hyperhidrosis, headaches (“characteristically unilateral and frontal”), edema (particularly in the infraorbital and malar areas), lacrimation, (hyper)salivation, and increased bladder tone (increased urinary frequency and enuresis) [2]. As we will defend in Section 4 (2nd hypothesis), after accepting that the SAFTS is indeed ADHD (1st hypothesis), is that ADHD is a systemic disease including both somatic and behavioral manifestations that may influence each other in a bidirectional manner (i.e., psyche influences soma, and soma influences psyche). Nowadays, there is strong and increasing evidence that patients with ADHD have a particular pattern of medical comorbidities.

### 2.3. A Misstep? Speer’s Cholinergic Theory

The first convincing hypothesis to explain SATFS in which children with atopic disorders developed hyperactivity and hyperesthetic behaviors was increased cholinergic activity at the cortical level triggered by the allergic stimulus, yet suggested by Speer [2,5].

In 1989, Marshall (“what are the biochemical or brain mechanisms involved in the pathogenesis of ADHD with an etiology of allergy?”) expanded Speer’s cholinergic hypothesis by suggesting that the relationship between allergies and ADHD may be explained by a cholinergic/adrenergic activity imbalance in the central nervous system (CNS) [6]. He stressed that the increased nasal and throat mucus and bronchospasm (respiratory system), pylorospasm, cramping, and constipation (digestive system) were symptoms typically associated with increased cholinergic tone. This cholinergic hyper-response, induced by an allergen, would lead to the predominance of cholinergic activity over adrenergic activity in genetically predisposed patients and explain both excitatory and attention deficit symptoms. Furthermore, this cholinergic–adrenergic imbalance would be potentiated, independently of the allergy, by all those stressors included in the patient’s environment. Some studies gave partial support to this hypothesis [8].

In any case, what is quite surprising is that although Speer was describing a syndrome characterized by allergies, he did not mention either immunoglobulin E (IgE) or histamine, which are probably the most relevant biomarkers of allergies. Indeed, we had to wait until 2009, when Pelsser and colleagues hypothesized that suggested the relationship between allergies and ADHD was supported by a hypersensitive mechanism (including both allergic—IgE- or non-IgE-mediated—and non-allergic mechanisms) [7]. In other words, this is the first time that IgE, and more indirectly, histamine, is mentioned as a plausible mechanism underlying the relationship between ADHD and allergies. See Figure 1.

IgE is an immunoglobulin that plays a crucial role in the immune response to allergens. When IgE binds to an allergen, it triggers the release of histamine and other mediators from mast cells, leading to an allergic response [9]. Furthermore, the initial symptomatic treatment of most allergic pathologies is antihistamines. However, what do we know about histamine?

## 3. Histamine

As we will defend later in point four (3rd hypothesis), histamine is critical for understanding the pathophysiology of both the behavior of ADHD and the typical medical comorbidities displayed by patients with ADHD. The human histaminergic system (histamine, an imidazole amine, and its four G protein-coupled receptors, HRs 1–4) is a complex system involved in basic physiological functions, such as the energy and endocrine homeostasis, digestion, sleep-wake cycle, cognition, attention, and immunoregulation, among others.

Although our body synthesizes histamine, foods also contain varying amounts of histamine. Furthermore, histamine has the potential to be toxic. The deleterious effects of excessive histamine ingestion were initially referred to as scombrotoxicosis (histamine intoxication or histamine poisoning) more than 60 years ago [10]. Histamine poisoning was related almost exclusively to the consumption of spoiled fish, particularly poorly transported tuna. It is characterized by occurring in outbreaks, having a short incubation period (around 20 min post-ingestion), and includes symptoms of low/moderate severity that remit in a few hours. The symptoms are closely linked to several physiological functions of histamine, affecting the gastrointestinal tract (e.g., nausea, vomiting, diarrhea), the skin (e.g., rash, redness, urticaria, pruritus, edema, local inflammation), and the cardiovascular (hypotension) and neurological (e.g., headache, palpitations) system [10].

In order to avoid potential intoxication or the deleterious effects of excessive histamine in the body, histamine is metabolized by two main enzymes: the DAO enzyme and histamine-N-methyltransferase (HNMT) [11,12]. HNMT is responsible for the degradation of intracellular histamine, whereas DAO metabolizes histamine extracellularly [10]. Whenever the activity of either DAO or HNMT is insufficient, histamine is accumulated extracellularly or intracellularly, respectively. Under physiological conditions, DAO has low activity in the brain and mainly catabolizes histamine in peripheral tissues. However, whenever the activity of HNMT is inhibited, DAO may help to catabolize brain histamine [13,14]. See Figure 2.

### 3.1. Body Histamine

Body histamine is mainly involved in local immune responses and the digestive system. The best-known histamine receptors, H1R and H2R, are low-affinity, classic drug targets for allergies and gastric ulcers, respectively. Lesser known but with high therapeutic potential, H3R and H4R “are high-affinity receptors in the brain and immune system, respectively” [15]. H1R is expressed in several cells (including mast cells) and is involved in Type 1 hypersensitivity reactions. H2R is mainly involved in Th1 lymphocyte cytokine production. H3R plays a role in the Blood–Brain Barrier (BBB) function. H4R is highly expressed in mast cells, and their stimulation increases histamine and cytokine production [16]. As I will stress in Section 4, all these diseases mediated by histamine are frequently comorbid conditions in patients diagnosed with ADHD.

#### 3.1.1. Histamine and the Immune System

Since its discovery in 1911, histamine has been involved in the pathophysiology of several somatic diseases. However, the role of body histamine is particularly relevant in regulating immunity. As a matter of fact, histamine has traditionally been recognized as a critical mediator in allergic diseases such as acute pruritus, atopic dermatitis, and allergic asthma Histamine is a potent inflammatory mediator with pleiotropic effects participating in the regulation of both innate and adaptive immunity [17]. For instance, histamine stimulates the recruitment of mast cells and basophils in the site of inflammation in response to various stimuli, including allergens, pathogens, and stress. By amplifying the inflammatory response, histamine may enhance the establishment of chronic inflammation. This pro-inflammatory effect appears to be mediated by H1R. In addition to this innate, pro-inflammatory immune response, histamine is also a key player in the regulation of adaptive immune responses. In other words, histamine mediates both the early- and late-phase reactions of an allergic response. To sum up, the pleiotropic effect of histamine allows it to exert broad effects, including inflammation and adaptive immune responses at a systemic level (i.e., vascular system, airways, intestine, microbiota, skin, and nervous system). See Figure 3.

Antihistamines have been used to treat allergies mediated by H1R. Unfortunately, the use of first-generation antihistamines was limited because they penetrate the BBB and cause sedation, among other side effects. Second-generation anti-H1R drugs (i.e., ebastine) do not penetrate the BBB and thus have improved tolerability [18]. H4R also plays a critical role in the development, progression, and modulation of many allergic diseases [16]. Accordingly, H4R antagonists might also be used as anti-allergy drugs as they have shown promising effects in preclinical and clinical studies in the treatment of several allergic diseases, inflammation, and autoimmune disorders [16].

#### 3.1.2. Histamine and the Digestive System

Regarding the digestive system, H2R antagonists are yet considered traditional treatments for a variety of gastric diseases, such as gastritis or gastro-esophageal reflux, given their gastroprotective effect. One of the major uses is the prevention or treatment of peptic ulcers or bleeding associated with non-steroidal anti-inflammatory drug (NSAIDs) use [19]. More recent research also suggests that H2R antagonists can also improve dysbiosis secondary to NSAID use [20].

#### 3.1.3. Histamine and Migraines

The use of drugs targeting histamine receptors is increasingly used in migraine. Histamine has a vasodilative effect. There is reduced susceptibility for migraine attacks during the evening, which corresponds with less central histaminergic firing [21]. Furthermore, a recent review stressed that histamine is critical in migraine pathogenesis via an inflammation pathway. The authors suggested that the activation of H3R lowers the release of histamine; drugs targeting this receptor may have anti-nociceptive and anti-neurogenic inflammatory actions that could be used for treating migraine [22].

#### 3.1.4. Histamine and Histamine Intolerance (HIT)

Here, it is interesting to include a less-known pathology, histamine intolerance (HIT). HIT may be defined as a non-immunological disease characterized by an impaired ability to metabolize ingested histamine. Excessive body histamine can be secondary to (1) the ingestion of histamine-rich food that surpasses the catabolic capacity of the subject; (2) the intake of alcohol or other drugs that either release histamine and/or block the DAO; and (3) a genetic predisposition to a deleterious capacity for catabolizing histamine (genetic DAO deficiency). All three reasons may derive from the same clinical picture, but only the last two reasons are related to HIT. HIT may include both gastrointestinal and non-gastrointestinal non-specific symptoms such as food intolerance, allergies, and mastocytosis, among others [11]. HIT symptoms are indeed similar to those of histamine intoxication [23,24] but less acute and severe, and more prolonged.

HIT symptoms are linked to the various physiological functions of histamine in the body, affecting the skin (e.g., flushing, redness, rash, urticaria, pruritus), the gastrointestinal tract (e.g., nausea, diarrhea, intolerance of histamine-rich food, and alcohol), the respiratory (e.g., rhino-conjunctival symptoms; asthma, particularly during exercise), the cardiovascular (hypotension, arrhythmia), the reproductive (e.g., dysmenorrhea), and neurological (e.g., chronic headache, migraine) bodily functions [25]. Recall that all these systems are exactly the same involved in SATFS. Unfortunately, the diagnosis of HIT is usually difficult because symptoms are usually unspecific and mild. Furthermore, the prevalence of HIT is probably underestimated, given the multifaceted nature and the similarity of the symptoms with allergic diseases and other disorders.

Currently, a positive response to a low-histamine diet is the most relevant for establishing the diagnosis. Similarly, following a low-histamine diet is the main therapeutic option for avoiding HIT symptoms [10]. However, what is the main ethological factor in HIT?

#### 3.1.5. DAO Deficiency and Histamine-Related Diseases, including Histamine Intolerance (HIT)

An impaired histamine degradation based on reduced DAO activity is the most frequent cause of HIT. DAO activity may be the result of genetic DAO deficiency, environmental DAO deficiency (secondary to the intake of alcohol and/or drugs inhibiting DAO), or both. DAO deficiency can lead to an excessive accumulation of histamine in the body, which has been linked to a number of symptoms and diseases, including (seasonal) allergic diseases [26] and asthma in some people [27]. Histamine can cause airway constriction, which can worsen asthma symptoms. Therefore, DAO deficiency may be a contributing factor in the development and severity of asthma. In a recent interesting study, reduced DAO serum levels leading to the occurrence of HIT symptoms were significantly more frequent in patients with atopic eczema than in controls [25]. DAO deficiency has also been related to HIT, and other food allergies/intolerance, particularly lactose intolerance [28]. In another study, the authors reported a very close relationship between DAO deficiency and non-celiac gluten sensitivity (NCGS), thus leading to the most severe migraine symptomatology, as both gluten and histamine are commonly involved within migraine [29]. Histamine present in food is one of the main triggers of food allergies, and DAO deficiency may increase susceptibility to these allergies. As stated before, histamine is an inflammatory biomarker involved in local immune responses [17]. Another typical medical condition associated with DAO deficiency is migraine. Indeed, more than 85% of patients affected by migraine display DAO deficiency [30]. Furthermore, genetic DAO deficiency appears to be very frequent in patients diagnosed with fibromyalgia [31]. Recall that, in Table 1, we included some symptoms described in the fatigue (motor) component of SATFS (tiring very rapidly and easily and complaints of muscular weakness and achiness), which are typical symptoms reported by patients with fibromyalgia.

There are several candidate tests to detect genetic DAO deficiency. However, their informative value is questionable, as the DAO expression and activity depend on the interplay between genes and the environment [12]. Furthermore, DAO has a higher expression than HNMT, being the main barrier for the intestinal absorption of histamine [32]. At least in the Caucasian population, the SNPs that can most directly cause DAO deficiency are rs10156191, rs1049742, rs2052129, and rs1049793 [12,31].

To treat DAO deficiency, dietary supplements containing the DAO enzyme have been developed. These supplements have been used successfully in reducing the symptoms of allergic diseases and asthma in some patients with DAO deficiency. However, food supplements should not replace a healthy, balanced diet. Indeed, a histamine-free diet is the treatment of choice for treating HIT [33].

In conclusion, DAO deficiency may be related to an increased susceptibility to allergic diseases and asthma, migraine, and several other medical comorbidities typically reported by patients diagnosed with ADHD. Excessive accumulation of histamine in the body can lead to more severe allergic reactions and worsen asthma symptoms. Dietary supplements containing the enzyme DAO may help reduce symptoms in some patients, but it is important to follow a healthy, balanced diet and seek medical attention if symptoms persist. Other therapeutic options, such as DAO supplementation, antihistamines, or probiotics, are considered complementary treatments.

### 3.2. Brain (CNS) Histamine

There is increasing evidence of the relevance of histamine in the CNS, where it acts as a neurotransmitter. In the brain, histamine is mainly produced by a group of tuberomammillary neurons concentrated in the posterior hypothalamus (TNM), which project practically to the whole CNS, and a limited number of mast cells throughout the CNS [15,21,34]. In mammals, these neurons emerge late and mature slowly [34,35]. CNS histaminergic neurons play a key role in the regulation of memory, learning, locomotion, circadian rhythms, and feeding, among others [36].

Both H1R and H2R are relevant postsynaptic receptors in the CNS, as they mediate some of the central effects of histamine, such as alertness and wakefulness. H3R is a pre- and postsynaptic receptor. H3R regulates the release of histamine and many other neurotransmitters. H4R is found in microglia and cerebral blood vessels. The expression of H4R in neurons is not yet well established [15]. See Table 2.

A recent review focused on the role of the histaminergic system in neuropsychiatric disorders [37]. Curiously, the authors reported strong evidence linking histamine with different neuropsychiatric disorders traditionally associated with the dopamine system (see Table 2), but there was not a single word about ADHD.

Furthermore, there are several treatments for CNS disorders targeting histaminergic receptors: H1R antagonists have been used to treat insomnia. Unfortunately, the use of first-generation H1R antagonists capable of crossing the BBB is limited by sedation. H2R antagonists have shown some efficacy in schizophrenia, but their use is limited nowadays. An H3R antagonist, Pitolisant, is used to treat hypersomnia and narcolepsy. Finally, H4R ligands may play a role in neuroimmunological disorders and neurodegenerative disorders, but clinical tests are lacking [15].

In theory, somatic (blood) histamine does not pass the BBB. However, during development, the BBB is permeable to histamine. Accordingly, DAO deficiency may influence not only a series of somatic alterations, such as gastric or allergic manifestations but also some manifestations in the brain, such as ADHD symptoms. Furthermore, as stated before, 85% of patients with migraine have DAO deficiency. Furthermore, “systemically given histamine may elicit, maintain, and aggravate headache” [21].

## 4. Speer Allergic Tension-Fatigue Syndrome (SATFS) Re-Visited

Here, I will defend the hypotheses stated in the introduction, namely:Hypothesis 1: SATFS is probably one of the first and best historical descriptions of patients with ADHD;Hypothesis 2: SATFS (ADHD for me) is a systemic disease;Hypothesis 3: Histamine is key for understanding the pathophysiology of the typical medical comorbidities displayed by patients with ADHD; andHypothesis 4: The DAO enzyme may play a key role in the pathophysiology of ADHD. A decreased DAO activity may lead to an accumulation of histamine, which could contribute to ADHD symptoms.

However, what is the evidence for these hypotheses?

### 4.1. Might the Behavior Displayed by Patients Diagnosed with SATFS Be an Initial Description of ADHD? (Hypothesis 1)

To our knowledge, there are no previous studies suggesting that SATFS is a good clinical description of some patients with ADHD. Speer [1,2] stressed that the tension-fatigue syndrome was the result of the combination of (1) a behavioral syndrome that he said was indeed a “primary allergic disorder of the nervous system”; and (2) a systemic (constitutional) syndrome (more than a disease) involving several apparatuses, but particularly, the allergic system.

As stated in Table 1, most of the behavioral symptoms included in SATFS are typical symptoms of ADHD. SATFS syndrome includes a typical behavior with (1) a motor component (hyperkinesis): exaggerated, accelerated, and continuous motor function; impatient, talkative, fidgety, poorly coordinated, and accident-prone; and (2) a sensory component (hyperesthesia): insomnia, irritability, distractibility, short attention span, excitability, and unusual sensitivity (oversensitivity) to innocuous stimuli (i.e., noise and temperature change, oversensitivity to pain) [1,2].

The hyperkinetic component of SATFS is clearly reminiscent of the hyperactivity component of ADHD, while the hypersensitive component clearly alludes to symptoms of the inattentive component of ADHD (“The essential characteristics of allergic hyperkinesis are exaggeration and acceleration of motor function. All muscular activity is likely to be jerky, quick, and overdone. […] hyperkinetic allergic children are often rather poorly coordinated, clumsy, and accident-prone. They are commonly deficient in manual skills, and although their school work is generally good, they often receive poor marks in writing, art work and crafts. There is evidence that their poor coordination may at times extend to speech function, with resultant stuttering and related problems[…]. Teachers find the child to have a short attention span and to be timid, restless, and distractible. Grades often suffer, and it is a commonplace comment of both parents and teachers that “he could learn if he would only pay attention”. Since these children neither understand themselves nor are understood by their elders, it is not uncommon to find them to be sullen and negativistic and convinced that the world is against them […]. But a large number of them remain retiring, oversensitive, insecure and unhappy—and unrecognized victims of constitutional allergy”) [2].

The fatigue sensory component of SATFS (being inactive, in a torpor, sluggish, sleepy, and apathetic state) clearly recalls either some inattentive symptoms or even the symptoms included within the concept of Sluggish Cognitive Tempo (SCT) [38]. There is still controversy as if SCT is a subtype of ADHD or a separate disorder. Furthermore, the unusual sensitivity to stimuli is more typically found in patients with autism spectrum disorder (ASD) but is also present in patients with ADHD. Taking into account all this evidence, why did Speer not allude to the concept of ADHD when describing SATFS?

The most plausible explanation is that, although several authors coined different terms for hyperactive children before 1950, including Franz Kramer and Hans Pollnow, it was not until the sixties, with the work by Stella Chess, that ADHD was nominated and began to become notorious in the scientific community [4]. Indeed, ADHD was first included in the third version of the Diagnostic and Statistical Manual of Mental Disorders (DSM-III) in 1980. Accordingly, it was virtually impossible for an expert on pediatric allergies, but not psychiatry, to notice that what he was describing was indeed an emerging neuropsychiatric disorder (ADHD) even poorly known by early alienists.

### 4.2. The Constitutional Component of SATFS: ADHD as a Systemic Disease. What Is the Evidence? (Hypothesis 2)

In their original studies, Speer stressed that the behaviors of allergic children “constitute a primary allergic disorder of the nervous system” [2]. In other words, he stressed that the behavior itself was part of a systemic (allergic) disease. After accepting Hypothesis 1 (SATFS corresponds to the clinical description of ADHD), which is the evidence about ADHD as a systemic disease? First of all, there is increasing literature suggesting that ADHD is a systemic inflammatory disease, in the same vein as depression was previously depicted [39]. People with ADHD frequently have comorbid disorders Recent evidence stemming from three ground-breaking studies curiously stressed similar comorbidities in the same somatic systems. In the first of these studies, the authors focused only on frequent disorders with well-documented large-scale genetic and epidemiological evidence for association with ADHD [40]. They grouped the comorbidities into three principal groups: (1) cardiometabolic (obesity, coronary heart disease, and type 2 diabetes); (2) immune–inflammatory–autoimmune (asthma, dermatitis, Crohn’s disease and ulcerative colitis, and rheumatoid arthritis); and (3) neuropsychiatric (migraine, insomnia, epilepsy, major depressive disorder, schizophrenia, alcohol intake and smoking, and autism spectrum disorder). Interestingly, the allergic and behavior symptoms included in SATFS are included within these comorbidities.

In the second study, the authors tested whether or not ADHD polygenic risk scores (PRS) were associated with mid-to-late-life somatic diseases in the general population [41]. They included 16 diseases particularly prevalent during middle age, and again, they grouped them into the same groups: (1) cardiometabolic (ischemic heart disease, heart failure, cerebrovascular disease, peripheral vascular disease, hypertension, obesity, and type-2 diabetes); (2) autoimmune–inflammatory diseases (type 1 diabetes, psoriasis, inflammatory bowel disease, and rheumatoid arthritis); and (3) neurological conditions (migraine, epilepsy, dementia, Parkinson’s disease and Parkinsonism, and sleep disorder). They found that higher ADHD–PRS was associated with increased risk of seven somatic conditions (that were, in order of relevance: type 1 diabetes, peripheral vascular disease, rheumatoid arthritis, obesity, heart failure, cerebrovascular disease, and migraine), thus concluding that ADHD conferred the risk for these comorbid outcomes, even in the absence of drug treatment for ADHD.

The third study focused on the phenotypic and etiological associations between ADHD and several somatic conditions across adulthood by doing a registered study with a final sample of nearly five million individuals in Sweden [42]. Stronger cross-disorder associations were found for the nervous system (sleep disorders, Parkinson’s disease, dementia, migraine, and epilepsy), respiratory (asthma, EPOC), musculoskeletal (back pain), and metabolic diseases, but also genitourinary, circulatory, gastrointestinal, and skin diseases. They found that ADHD was associated with 34 out of the 35 (97%) conditions studied. Migraine showed almost complete genetic overlap with ADHD. The authors stressed the relevance of assessing the presence of physical diseases in ADHD patients.

In conclusion, all these studies suggest that ADHD is particularly comorbid with other neuropsychiatric, cardiometabolic, and immune–inflammatory–autoimmune conditions. Psychiatric comorbidities that are commonly associated with ADHD include sleep disorders, such as obstructive sleep apnea and restless legs syndrome [43], depression and anxiety disorders [44], and substance use disorders [45]. Regarding comorbid cardiovascular disorders, several studies have demonstrated a significant association between ADHD and obesity in both children and adults [46]. However, the immune–inflammatory–autoimmune conditions were the ones more clearly depicted in the initial description of SATFS. Accordingly, we expand on this relationship.

#### ADHD and the Immune System: A Focus on Inflammation, Allergy, and Autoimmune, Cancer, and Cardiovascular Comorbidities

Generally speaking, the immune response can be defined as the body’s ability to stay safe by affording protection against harmful agents. The immune system plays a critical role in the body’s response to infection, injury, and disease. Immunity includes both a first-line line of defense, innate immunity (inflammation: inflammatory cells that trap bacteria and other offending agents and start healing injured tissue), and highly specific responses to a particular offender (adaptive immunity). However, inflammation can be harmful if it becomes chronic or dysregulated [47]. Indeed, chronic inflammation has been associated with a wide range of diseases, typically allergy and autoimmune disorders, and more recently, cancer and cardiovascular disease [17]. In these conditions, the immune system may become overactive and mistakenly target healthy tissue, leading to chronic inflammation and tissue damage.

All the allergic (atopic) symptoms included in SATFS are frequently reported by people diagnosed with ADHD. The first authors to draw attention to the high prevalence of autoimmune and atopic diseases in patients with ADHD were Geschwind (1985) [48,49] and Marshall (1989) [6]. Geschwind proposed a causal relationship between non-right-handedness, immune disorders, ADHD, and learning disabilities via prenatal exposure to high levels of testosterone, whose effects would involve alterations of the immune system and reading and motor alterations at the CNS [48,49,50]. Later on, Marshall (1989) carried out an exhaustive study of the “allergic tension-fatigue syndrome” [6]. He expanded Speer’s cholinergic hypothesis and suggested that the bridge between ADHD and allergies was a cholinergic–adrenergic imbalance.

Since the 1990s, several studies have emerged testing the association between atopia and ADHD. Regarding the relationship between atopic dermatitis and ADHD, most authors reported an association [51,52,53,54,55,56,57,58], but some did not [59,60,61]. Again, regarding the relationship between asthma and ADHD, most authors report an association [57,58,62,63,64,65,66,67], but some do not [59,68,69]. Finally, regarding the relationship between allergic rhinitis and ADHD, most studies report a positive relationship [53,54,57,58,59,70,71]. In any case, two recent systematic reviews and meta-analyses provided convincing evidence that ADHD is related to atopic diseases [72,73]. An even more recent study integrating information from two meta-analyses concluded that atopic diseases were not only associated with ADHD but also with ADHD symptom severity [74].

Furthermore, patients with ADHD had a higher prevalence of autoimmune diseases, including type 1 diabetes, rheumatoid arthritis, and inflammatory bowel disease [40,41,75]. Chronic inflammation and immune dysregulation may probably play a role in the pathophysiology of both conditions. A personal history and a maternal (but not paternal) history of autoimmune disease has been associated with an increased risk of ADHD [76].

Although the study of the relationship between cancer and ADHD is still in its infancy, there are recent interesting studies linking both pathologies. For instance, a retrospective cohort study carried out in Taiwan reported that ADHD was associated with an increased risk of colorectal cancer (adjusted Hazard Ratio = 3.458, 95% CI = 1.640–7.293, *p* < 0.001) [77]. Furthermore, a history of a neurodevelopmental disorder (ADHD, autism spectrum disorder, and intellectual disabilities) was associated with an increased risk of seminoma) in a nested case-control study (OR: 1.54; 1.09–2.19 [78]. Moreover, ADHD shows substantial genetic correlations with nonpsychiatric conditions, including lung cancer [79]. However, the most relevant study on the relationship between ADHD and cancer for the present work is a recent study revealing that cancer patients who took antihistamines during immunotherapy had significantly improved survival. This observation led this team to find that HR1 was frequently increased in the tumor microenvironment, thus inducing T-cell dysfunction. Furthermore, antihistamine treatment restored immunotherapy response by reverting macrophage immunosuppression and revitalizing T cell cytotoxic function. In other words, clinical manifestations of allergies via H1R facilitated both tumor growth and induced immunotherapy resistance. Thus, high histamine levels or pre-existing allergies in cancer patients may dampen immunotherapy responses. Accordingly, antihistamines might be used as adjuvant agents for combinatorial immunotherapy [80].

Regarding the relationship between the immune system (including inflammation) and cardiovascular diseases, a recent prospective cohort study including 414,495 participants demonstrated a relationship between a total of 20 individual immune-mediated diseases and an increased risk of incident cardiovascular disease [81]. Furthermore, trained immunity (innate immune memory), a persistent hyper-responsive functional state of innate immune cells, is associated with an underlying mechanism of chronic inflammation in atherosclerotic cardiovascular disease [82]. Moreover, existing studies postulate a crucial role for inflammation and inflammatory cells, including mast cells, in cardiovascular diseases. Histamine, the critical mast cell mediator, and its receptors profoundly impact the pathophysiology of cardiovascular diseases such as hypertension-induced cardiac hypertrophy. Several preclinical and clinical studies using histamine receptor antagonists, particularly H2Rs, report improvement in cardiac function, thus paving the way for repurposing antihistamines for cardiovascular diseases [83].

In conclusion, there is evidence to suggest a link between ADHD and immune-related diseases. The relationship between immunity and inflammation is complex and multifaceted, but a well-functioning immune system and balanced inflammatory response are essential for maintaining health and preventing disease [84]. The role of histamine, involved in both innate (inflammation) and adaptive immune responses, is probably critical.

### 4.3. Is Histamine the Missing Link between ADHD and Allergies? (Hypothesis 3)

After accepting Hypothesis 1 and 2, why histamine? As stated before, ADHD is a systemic condition in which the role of the immune system, particularly inflammation, is central. Histamine is a potent inflammatory mediator with pleiotropic effects derived from its activity through four different coupled protein receptors, which produces a fine-tuned regulation of the different immune cells, participating in the regulation of both innate and adaptive immunity, including the production and release of cytokines [85]. Histamine is frequently associated with allergic reactions, which stimulate the migration of eosinophils from the bloodstream, and the recruitment of mast cells in the site of inflammation. By amplifying the inflammatory response, histamine may enhance the establishment of chronic inflammation. In addition to this innate, pro-inflammatory immune response, histamine is also a key player in the regulation of adaptive immune responses. To sum up, the pleiotropic effect of histamine allows it to exert broad effects, including inflammation, adaptive immune responses, and regulation at a systemic level (i.e., vascular system, airways, intestine, microbiota, skin, and nervous system) [17]. In any case, further research is needed to fully understand the relationship between histamine and cytokines and its implications for health and disease.

Here, we also want to emphasize that by supporting the role of histamine on ADHD, we do not want to downplay the importance of the cholinergic system and the role of neurotransmitters such as dopamine or noradrenaline, which have a fundamental role in the pathophysiology of ADHD. We simply want to emphasize that histamine may be the missing link capable of explaining some aspects of the behavior, particularly the somatic comorbidities typically displayed by people with ADHD. In this respect, we again stress that it is awkward that, given the close relationship between ADHD and allergies, there is virtually no literature suggesting that histamine may play a role. In this section, we will review several lines of evidence suggesting a relationship between ADHD and histamine.

#### 4.3.1. A Little Bit of History—The Physiopathological Foundations of ADHD

As stated before, in 1989, Marshall postulated that a cholinergic/adrenergic activity imbalance in the CNS was critical to explain the relationship between ADHD and allergies [6]. Acetylcholine is an important neurotransmitter that plays a crucial role in the regulation of the parasympathetic nervous system. It is involved in many physiological processes, such as muscle contraction, cognitive function, and regulation of the immune system.

In 2009, Pelsser and colleagues suggested that the relationship between allergies and ADHD was supported by a hypersensitive mechanism, including both allergic (IgE- or non-IgE-mediated) and non-allergic alternatives [7]. These authors considered ADHD a hypersensitivity syndrome with(out) allergic basis, in which dopamine, noradrenaline, and histamine would play a common role in the ADHD–asthma duet, causing exaggerated reactions to certain environmental stimuli in both pathologies. This is the first time that histamine is postulated as a neurotransmitter potentially involved in the ADHD–atopia relationship.

#### 4.3.2. ADHD and Histamine-Mediated Comorbid Disorders

Although the step taken by Pelsser et al. was relevant, they did not take a final step to relate IgE to histamine. In other words, a potential exploratory model including inflammation remained unexplored. As a matter of fact, recent research suggests that inflammation is key in the pathogenesis of ADHD [86]. ADHD is strongly associated with several allergic and autoimmune diseases, such as asthma, eczema, urticaria, and ulcerative colitis [40]. All these disorders can be found in the initial descriptions of SATFS, and in all of them, histamine plays a major role. As said before, histamine is one major player in inflammation and allergies, stimulating the release of cytokines that increase the release of dopamine and may also modulate brain maturation. Delay in brain maturation is typically found in people with ADHD, and comorbid neurodevelopmental disorders, such as learning disorders [87]. Furthermore, the Geschwind-Behan hypothesis study reported an association between left-handedness, which is more frequent in patients with ADHD, and immune disease, migraine, and developmental learning disorder [88]. Interestingly, in 1971, Speer published a lesser-known paper also relating to allergies and migraines [89]. Recent studies have given some support to Geshwind’s hypothesis of a relationship between cerebral dominance and a characteristic immunologic set [90,91]. In the most relevant study to date, the authors reported that individuals indicating “either” hand for writing preference “had significantly lower spatial performance (mental rotation task) and significantly higher prevalence of hyperactivity, dyslexia, asthma than individuals who had clear left- or right-hand preferences” [92].

Regarding the relationship between left-handedness and migraine, the evidence is more controversial [93,94,95]. In any case, it is interesting to stress the results of a study in which the authors reported a close relationship between the number of anomalous brain conditions or phenomena (including mixed- or left-handedness, learning and speech disorders, enuresis after age 5), and migraine [96].

Moreover, the evidence regarding an association between left-handedness and both dyslexia and ADHD is compelling [97,98,99,100]. A recent meta-analysis confirmed the relationship between left-handedness and ADHD [101]. Furthermore, left-handedness has also been related to different situations associated with ADHD, such as developmental coordination disorder [102], traumatic dental injuries [103], head injuries [104], and enuresis [105]. Furthermore, neonatal habenula lesion-induced ADHD-like syndrome in juvenile rats could be normalized by the histamine H3 receptor antagonists [106].

#### 4.3.3. ADHD and Histamine: What Is the Evidence?

As stated before, histamine is a biogenic amine that acts as a neurotransmitter in the CNS and is involved in a wide range of physiological and pathological processes. Histamine is synthesized in the brain and modulates several neurotransmitter systems, including the catecholaminergic system (dopamine and noradrenaline), which is the major neurotransmitter system associated with the pathophysiology of ADHD. Dopamine is a neurotransmitter that plays a crucial role in the regulation of the reward system, motivation, movement, and immune system [107]. Noradrenaline is involved in the regulation of the sympathetic nervous system and plays a crucial role in the regulation of the “fight or flight” response and the regulation of the immune system [108]. Interestingly, all major drugs (i.e., atomoxetine, lisdexamfetamine, methylphenidate) treating ADHD increase either dopamine or noradrenaline, or both in the synapsis cleft. Interestingly, all these drugs also increase histamine in the brain.

Furthermore, given the widespread use of antihistamines in allergic diseases and the well-known deleterious effect of traditional antihistamines, capable of crossing the BBB on attention and concentration, one might expect that the role of histamine on ADHD has extensively been explored, but this is not the case. The following data suggest that histamine is closely associated with ADHD and comorbid disorders.

(1)One of the most relevant lines of evidence comes from the deleterious effects produced by some antihistaminergic drugs. Traditionally, first-generation antihistamines (i.e., diphenhydramine) were characterized by readily crossing the BBB, thus leading to significant CNS side effects such as altered mood, reduced wakefulness, drowsiness, and impaired psychomotor and cognitive performance (i.e., vigilance, divided attention, and working memory) even in the absence of self-reported sleepiness [109]. In other words, antihistamines produce side effects that are very similar to some ADHD symptoms.(2)Histamine levels [37,110], serum DAO levels [111], and HNMT SNP variants [112] are associated with the development of cognitive and neuropsychiatric disorders, including ADHD.(3)Two frequently used drugs for treating ADHD (methylphenidate and atomoxetine) increase histamine availability in the pre-frontal cortex [113,114]. Thus, the therapeutic effects of ADHD medications may partly be due to increasing histamine release, in addition to the well-known increasing effect of dopamine and noradrenaline in ADHD [115]. A third drug, lisdexamfetamine dimesylate, an amphetamine approved for the treatment of ADHD, promoted a strong upregulation of DAO mRNA levels, suggesting that this drug may induce DAO activity [116], thus helping to decrease blood histamine levels. This finding is interesting because, with a single drug, we may “kill two birds with one shot”, as we may be treating not only core ADHD symptoms but also allergies.(4)Antihistamine use has been associated with subsequent detection of ADHD [117,118,119]. This increased use of antihistamine drugs may be related to the increased risk of diseases such as atopy [119,120], food allergies [121], and allergic rhinitis [122] in people with ADHD.(5)There is increasing evidence concerning the potential therapeutic use of drugs acting on the histaminergic system in patients diagnosed with ADHD. In 1985, a case study reported that an antihistamine, anti-motion sickness drug might exert some improvement in ADHD [123]. While the precise mechanisms underlying the relationship between histamine and ADHD are still unclear, several preclinical and clinical studies have suggested that H3R antagonists, such as Pitolisant, may be effective in treating ADHD symptoms. These drugs increase histamine release and have been shown to improve cognitive function [124] and reduce hyperactivity in individuals with ADHD [106]. However, some studies using anti-H3R drugs for the treatment of ADHD have yielded negative results [125,126].

#### 4.3.4. The Interaction between Histaminergic and Acetylcholine Systems

As stated previously, by stressing the role of histamine in ADHD, we do not want to downplay the role of the acetylcholine system. Both acetylcholine and histaminergic systems are important neurotransmitters that play a role in many physiological processes in the body. The interaction between them is complex, bi-directional, and involves multiple receptors and signaling pathways. Histamine mediates allergic responses and inflammation, but it is also involved in the regulation of neurotransmission. Histamine neurons project to many brain regions and can modulate the release of acetylcholine [12,14]. Indeed, histaminergic neurons increase the release of acetylcholine mediated by dopamine [127]. Furthermore, lisdexamfetamine markedly increased acetylcholine efflux in the pre-frontal cortex and histamine efflux in both the pre-frontal cortex and hippocampus, thus suggesting that both acetylcholine and histamine might be involved in its therapeutic effects in patients with ADHD [128]. Reversely, acetylcholine also regulates histamine release in various tissues, including the brain. Indeed, the nervous system regulates the inflammatory response in real-time thanks to acetylcholine, in the same vein as it controls heart rate [129].

Furthermore, several studies have shown that histamine and acetylcholine interact in many physiological processes, including sleep, cognitive function, and inflammation. For example, histamine and acetylcholine act together to regulate the sleep-wake cycle, with histamine promoting wakefulness and acetylcholine promoting REM sleep [36]. In summary, histamine and acetylcholine are important neurotransmitters that play a role in many physiological processes and may play a role in the pathophysiology of ADHD.

### 4.4. Might DAO Enzyme Deficiency Play a Role in the Pathophysiology of Core ADHD Symptoms and Somatic Comorbidities (Hypothesis 4)?

Our last hypothesis should be understood as a corollary of the third hypothesis. It is probably the most disruptive of the four hypotheses, but it is also the most difficult to defend due to the scarcity of existing literature, mostly indirectly linking DAO activity and ADHD. What we defend here is that the DAO enzyme is probably closely related to ADHD symptoms and the somatic diseases frequently associated with ADHD.

One of the major causes of an accumulation of histamine is decreased DAO activity. As stated before, SATFS included somatic symptoms that are frequently reported in patients with ADHD, but also in patients with either HIT or DAO deficiency (See Table 3): (1) the respiratory system (asthma, nasal allergy); (2) the digestive system (gastrointestinal allergy); (3) the skin (eczema, edema); (4) the CNS (headache, with a clinical description that recalls migraine, characteristically unilateral and frontal, typically associated with HIT); and (5) the urinary system (enuresis).

For instance, there is an increased risk of migraine among patients with ADHD patients [130,131]. Interestingly, genetic DAO deficiency is related to migraine [132]. Indeed, more than 85% of patients diagnosed with migraine display DAO deficiency [30]. Moreover, ADHD is closely associated with celiac disease [133] and other food allergies that might be the counterpart of what Speer coined as “gastrointestinal allergy” within SATFS. Indeed, Tryphonas and Trites (1979) reported that 47% of the 90 hyperactive children referred were allergic to at least one food [134]. Again, Furthermore, a decreased DAO activity is associated with celiac disease, and gluten intolerance [29]. In addition, Maintz and colleagues proposed that diminished histamine degradation secondary to reduced DAO activity may be the main reason for non-IgE-mediated food intolerance caused by histamine [25].

Furthermore, the potential role of food additives in ADHD has long been debated, and the evidence appears to suggest such a relationship [135]. This is interesting because the exacerbation of ADHD symptoms in children by food additives seems to be related to a genetic polymorphism affecting histamine [136]. In this double-blind, placebo-controlled crossover trial, the adverse effect of food additives on ADHD symptoms was moderated by two HNMT polymorphisms and one DAT1 polymorphism. Moreover, HIT, a non-immune reaction caused by histamine accumulation—in some instances caused by DAO deficiency—is characterized by diverse skin, respiratory, and gastrointestinal pseudoallergic symptoms [137].

Finally, to our knowledge, there is not a single study addressing a direct relationship between DAO, the major catabolizing enzyme of histamine, and ADHD. If our hypothesis is that DAO deficiency is critically associated with ADHD, the prevalence of SNPs related to genetic DAO deficiency among patients diagnosed with ADHD should be high. As a matter of fact, in a pilot study, we have reported that nearly 80% of children and adolescents diagnosed with ADHD had a genetic DAO deficit, and in nearly 20% this deficit was severe (https://www.aepap.org/sites/default/files/documento/archivos-adjuntos/878-texto_del_articulo-3110-2-10-20220623.pdf accessed on 1 Augutst 2023). Currently, we are writing a paper based on this set of data of 300 children and adolescents with ADHD.

## 5. Conclusions, Limitations, and Future Directions

In this conceptual paper, we have presented compelling evidence suggesting that (1) the SAFTS is indeed an initial description of the emerging concept of ADHD in the sixties. Furthermore, we have explained why Speer could have never suspected that he was describing an emerging psychiatric disorder, ADHD; (2) ADHD is a systemic disease in which the somatic comorbidities, yet described by Speer, are mainly affecting some specific body systems (particularly, the immune system—including allergies, autoimmune disorders, and cancer—the digestive system; the cardiovascular system; the CNS (migraine); and the urinary/reproductive systems); (3) The role of histamine into the pathogenesis of ADHD has surprisingly been neglected. I suggested that histamine is critical to explain the overrepresentation of allergies in patients with ADHD; and (4) DAO deficiency may also be a critical factor in explaining both core ADHD symptoms and histamine-mediated (allergic-immune basis) somatic comorbid disorders.

The major limitation of the present study is its speculative nature. However, conceptual papers are important to attract the attention of the scientific community in neglected areas of research. Furthermore, I think that I have supported each of the hypotheses with scientifically sound literature. Another limitation is that I focused on the DAO enzyme deficiency and did not even hypothesize if the HNMT enzyme might also be involved in the pathogenesis of ADHD. As a matter of fact, a couple of articles stressed that the inhibition of HNMT may be used in the treatment of ADHD [112,138]. I took this decision because (1) the literature on HNMT and ADHD is very poor; (2) the DAO enzyme is the major catabolizing enzyme of histamine and is more scattered than the HNMT, thus providing a more compelling basis to explain core ADHD symptoms and somatic comorbid conditions; and (3) we are currently finishing a study devoted to the relationship between four DAO enzyme SNPs and ADHD, and have first-hand information about this relationship. However, unfortunately, the study did not include any SNP of the HNMT enzyme.

Overall, the relationship between histamine and ADHD is complex and multifaceted, and further research is needed to fully understand the mechanisms underlying this connection. Nevertheless, these preliminary findings highlight the potential importance of histamine-signaling in the pathophysiology of ADHD and suggest that histamine-based therapeutics may represent a promising avenue for the treatment of this systemic disorder. Future research is needed to fully understand the relationship between histamine, the DAO enzyme, and other major players in the pathophysiology of ADHD.

## Figures and Tables

**Figure 1 jcm-12-05350-f001:**
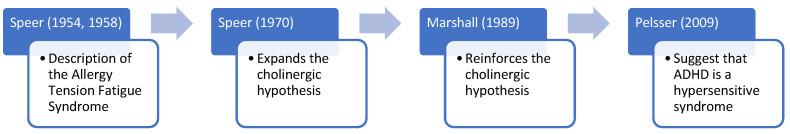
The relationship between allergies and ADHD: From Speer to the hypersensitive hypothesis of ADHD [1,2,5,6,7].

**Figure 2 jcm-12-05350-f002:**
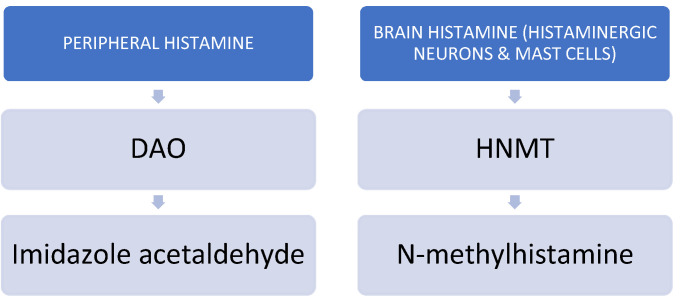
Main catabolic enzymes of histamine.

**Figure 3 jcm-12-05350-f003:**
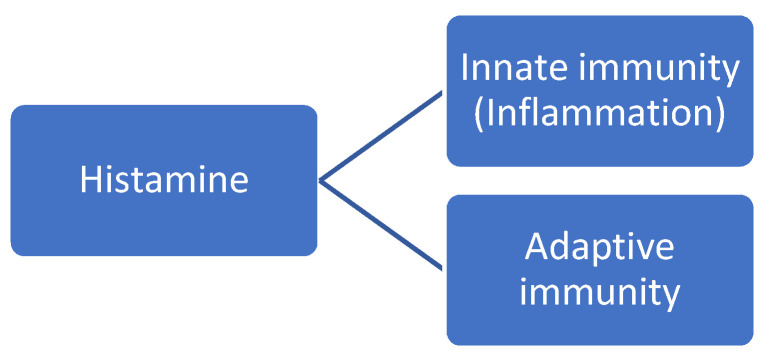
Roles of body histamine on the immune system.

**Table 1 jcm-12-05350-t001:** Summary of the behavioral symptoms of the allergic tension-fatigue syndrome.

Tension		Fatigue	
Motor	Sensory	Motor	Sensory
Exaggerated, accelerated, and continuous motor function	Unusual sensitivity (oversensitivity) to innocuous stimuli (i.e., noise and temperature change, oversensitivity to pain)	Tiring very rapidly and easily	Being in an inactive, torpor, sluggish, sleepy, and apathetic
Impatient	Irritability	Complaints of muscular weakness and achiness	
Talkative	Distractibility		
Fidgety	Short attention span		
Poorly coordinated	Excitability		
Accident prone	Insomnia		

**Table 2 jcm-12-05350-t002:** Histamine receptors in the CNS. Partially based on The Histaminergic System in Neuropsychiatric Disorders, published by Li Cheng, Jiaying Liu and Zhong Chen [37].

Histamine Receptor	Area and Cells	Function	CNS Pathology Involved
H_1_	Cerebral cortex, thalamus, locus coeruleus, raphe nucleus, astrocytes	Arousal and sleep-wakefulness	Narcolepsy Schizophrenia Alzheimer’s Disease
H_2_	Cerebral cortex, basal ganglia, hipoccampus, amygdala	Motoric	Schizophrenia Parkinson’s Disease Tourette’s syndrome
H_3_	All CNS	Regulates the release of other neurotransmitters, such as glutamate, GABA, acetylcholine, and dopamine both in CNS and periphery	Schizophrenia Alzheimer’s Disease Parkinson’s Disease Tourette’s síndrome ADHD
H_4_	Hematopoietic and immune cells (eosinophils, mast cells, and dendritic cells)	Inflammatory processes	

**Table 3 jcm-12-05350-t003:** Comparison between SATFS syndrome, systemic diseases frequently comorbid with ADHD, and HIT/DAO deficiency.

SATFS	Somatic Diseases Associated with ADHD	HIT and/or DAO Deficiency (Histamine Receptors)
**Gastrointestinal**	**Gastrointestinal**	**Gastrointestinal (H_1_/H_2_)**
		Bloating
		Postprandial fullness
Diarrhea		Diarrhea/Loose stool
Abdominal pain	Ulcer or chronic gastritis	Abdominal pain
Constipation		Constipation
Colic	Acute appendicitis	Intestinal colic
		Belching
Vomiting		Nausea/Vomiting
Food and drug sensitivities (Gastrointestinal allergies)	Celiac disease and other food allergies	Food sensitivities and allergies
Increased salivation		
	Fatty and alcohol liver disease, gallstone disease	
	Inflammatory bowel disease	Chrohn/colitis
**Skin**	**Skin**	**Skin (H_1_)**
	Psoriasis	Pruritus
Eczema	Atopia (Eczema)	Rash, eczema, urticaria
Edema (infraorbital) Deep red discoloration under the eyes		Swollen, reddened eyelids Flush
Hyperhidrosis, lacrimation		Excessive sudden sweating
		Vitiligo (also autoimmune)
**Nervous system**	**Nervous system**	**Nervous system (H_3_)**
Sluggish, torpor	Sleep disorders	Dizziness
Headache (unilateral)	Headache (including Migraine)	Headache (Migraine)
Nightmares, bad sleep	Sleep problems (insomnia)	Insomnia, circadian rhythm problems, obstructive apnea
	Epilepsy	
Delirium	Dementia and Parkinson’s	
**Circulatory system**	**Circulatory system**	**Circulatory system (H_1_/H_2_)**
		Tachycardia, palpitations, and/or arrhythmia
	Peripheral vascular disease, hypertension	Hypotension
	Heart failure, ischemic heart disease, hypertension, stroke	Collapse
**Respiratory system**	**Respiratory system**	**Respiratory system (H_1_)**
Blocked nose	Allergic rhinitis	Allergies (Rhinorrhea, nose congestion, sneezing)
Rapid fatigue	Chronic obstructive pulmonary disease (COPD)	Dyspnea, chronic coughing
Asthma	Asthma	Asthma
**Musculoskeletal**	**Musculoskeletal**	**Musculoskeletal**
Vague, widespread aching	Fibromyalgia	Fibromyalgia
Muscular weakness	Arthrosis, dorsalgia, rheumatoid arthritis	Muscle and joint pain
**Genitourinary**	**Genitourinary**	**Genitourinary (H_1_/H_2_)**
Enuresis, increased bladder tone	Enuresis	Enuresis, overactive bladder
Urinary infection	Urolithiasis and kidney infections	
	Glomerular disease	Hypersensitivity to NSAIDs
		Dysmenorrhea, endometriosis
**Endocrine, metabolic or autoimmune**	**Endocrine, metabolic or autoimmune**	**Endocrine, metabolic or autoimmune**
	Type 1 and 2 diabetes	
	Obesity	
	Thyroid disorders	

## Data Availability

Not applicable.

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
