# Peer review of "Is Histamine and Not Acetylcholine the Missing Link between ADHD and Allergies? Speer Allergic Tension Fatigue Syndrome Re-Visited"

_jcm, 2023, doi:10.3390/jcm12165350_

Round 1

Reviewer 1 Report

Thank you for the opportunity to review : Is histamine, and not acetyl-choline, the missing link between ADHD and allergy? The Speer allergic tension fatigue syndrome re-visited.

This is manuscript is a concept paper which uses the medical literature to propose and support the hypothesis that histamine is the neurotransmitter responsible for the relationship between ADHD and allergy, and specifically a deficit in diamine oxidase, which metabolizes histamine.

The author presents a convincing and well-cited argument to support his hypothesis.

While the English is quite good, the manuscript would benefit from additional proof reading by a native speaker. I started to offer suggestions, but needed to get this review submitted, so here is a start to a more detailed assessment of the language: 

Line 56/57: the word ”Allergy” doesn’t seem to belong or punctuation is needed

Line 57: (ADHD for us) – can you explain this

Line: 88 add “Speer” instead of “he”

Omit “Please”

Line 141: add “the” before brain

Author Response

Thank you for the opportunity to review : Is histamine, and not acetyl-choline, the missing link between ADHD and allergy? The Speer allergic tension fatigue syndrome re-visited.

This is manuscript is a concept paper which uses the medical literature to propose and support the hypothesis that histamine is the neurotransmitter responsible for the relationship between ADHD and allergy, and specifically a deficit in diamine oxidase, which metabolizes histamine.

The author presents a convincing and well-cited argument to support his hypothesis.

While the English is quite good, the manuscript would benefit from additional proof reading by a native speaker.

RE: Thanks for your words. I have asked again my native American speaker and usual collaborator, Lorraine Maw, for further English proof correction.

I started to offer suggestions, but needed to get this review submitted, so here is a start to a more detailed assessment of the language: 

Line 56/57: the word ”Allergy” doesn’t seem to belong or punctuation is needed.

RE: The reviewer is right. Corrected.

Line 57: (ADHD for us) – can you explain this

RE: The reviewer is right. Corrected.

Line: 88 add “Speer” instead of “he”

RE: Corrected.

Omit “Please”

RE: Done.

Line 141: add “the” before brain

RE: Done.

Reviewer 2 Report

Dear Editor,

I trust this message finds you well. I am writing to you in regards to the manuscript recently submitted to the Journal of Clinical Medicine on the concept of Speer Allergic Tension-Fatigue Syndrome (SATFS), its correlation with Attention Deficit Hyperactivity Disorder (ADHD), and the proposed allergic pathophysiology.

While the premise of the research holds potential, I found that several aspects could benefit from revisions to provide a more lucid and substantial exploration of the topic. Here are some of the primary concerns and recommendations:

  1. The manuscript lacks a clear definition and delineation of SATFS, its symptoms, and triggers. This ambiguity complicates distinguishing it from ADHD and allergic reactions.
  2. The paper draws parallels between SATFS and ADHD, but the supporting evidence presented is insufficient, relying more on speculation than empirical data.
  3. The assertion that SATFS is a systemic disease stemming from allergic pathophysiology lacks robust evidence. Substantiating this claim necessitates more research citations and clinical trials.
  4. The introduction would benefit from a more thorough review of prior research and original studies by Speer instead of relying primarily on the authors' interpretations.
  5. The paper neglects to provide a comprehensive background on the role of histamine in the body and brain, an essential prerequisite to understanding the later arguments.
  6. A discussion about the lack of conclusive proof regarding SATFS or the cholinergic imbalance is absent, which could lead to reader confusion.
  7. The paper's complexity, in terms of medical jargon and advanced concepts, might impede comprehension for the general reader. A simpler presentation is recommended.
  8. A clear, layman's definition of the immune response and inflammation would aid reader understanding, given their critical role in the discourse.
  9. A clear problem statement or research gap needs to be articulated earlier in the introduction.
  10. The connection between histamine, immune response, and inflammation is not clearly established. Clarification or simplification would enhance reader comprehension.
  11. The correlation between immune response, inflammation, and ADHD, although apparent as the study focus, is not stated until later in the introduction, which disrupts the structure.
  12. The introduction is laden with intricate biochemical relationship details, which might overwhelm a general audience. Relegating these to the methods or results section might help.
  13. Diseases like cancer, cardiovascular disease, and autoimmune disorders are mentioned but not expounded upon, leaving readers unsure of their relevance to the study.
  14. The significance of Speer syndrome is relegated to the introduction's end. If crucial to the research, it should be highlighted earlier with a more clear definition of its role.
  15. The hypothesis is not explicitly stated in the introduction, which might leave readers adrift.
  16. The discussion includes an array of neurotransmitters and molecules, which might confuse readers. Greater focus on the role of histamine would be beneficial.
  17. The conclusion does not adequately summarize the study's findings, leaving the reader without a clear takeaway.
  18. The paper concludes with a strong statement on the dearth of studies exploring histamine or DAO, and ADHD, which seems contradictory to the prior discussion of histamine's potential role.
  19. The link between SATFS and ADHD is not fully explored in the discussion, which could confuse the readers.
  20. The claim that ADHD is primarily caused by an immune response to allergens lacks supporting evidence, creating a credibility issue.
  21. The paper lacks a comparative analysis with existing literature, which is essential to put the findings in context.
  22. The conclusion fails to reiterate the significance of the study, leaving readers uncertain of the implications.
  23. The paper would benefit from a more compelling concluding statement, encapsulating the research and its potential implications.

In light of these points, I recommend a thorough review and revision of the manuscript to present a more compelling, organized, and substantiated discussion. Thank you for considering these comments.

Kind Regards,

Moderate editing of English language required

Author Response

Dear Editor,

I trust this message finds you well. I am writing to you in regards to the manuscript recently submitted to the Journal of Clinical Medicine on the concept of Speer Allergic Tension-Fatigue Syndrome (SATFS), its correlation with Attention Deficit Hyperactivity Disorder (ADHD), and the proposed allergic pathophysiology. While the premise of the research holds potential, I found that several aspects could benefit from revisions to provide a more lucid and substantial exploration of the topic.

RE: Thanks for these words.

Here are some of the primary concerns and recommendations:

  1. The manuscript lacks a clear definition and delineation of SATFS, its symptoms, and triggers. This ambiguity complicates distinguishing it from ADHD and allergic reactions.

RE: The reviewer is right. Accordingly, now we have included the definition of the SATFS, symptoms, and triggers as explained in the seminal papers by Frederic Speer in 1954 and 1958.

  1. The paper draws parallels between SATFS and ADHD, but the supporting evidence presented is insufficient, relying more on speculation than empirical data.

RE: The reviewer is right but the problem is that there is virtually no information regarding the SATFS: https://pubmed.ncbi.nlm.nih.gov/?term=%22allergic+tension-fatigue+syndrome%22&sort=date

However, he/she is right in that I have to stress this. Accordingly, I have included some limitations, and the speculative nature of the present hypothesis paper (point 5, Conclusions, limitations, and future directions).

2. The assertion that SATFS is a systemic disease stemming from allergic pathophysiology lacks robust evidence. Substantiating this claim necessitates more research citations and clinical trials.

RE: The reviewer is right. I did not want to mean that SATFS (ADHD) stems from allergic pathophysiology, but that allergy is part of the syndrome. Indeed, the authors who suggested this in their paper ADHD as a (non) allergic hypersensitivity disorder: a hypothesis, were Pessler et al. in 2009. What I wanted to state is the increasing, and very sound literature postulating that ADHD is a systemic disease encompassing both a behavioral and a somatic component. Accordingly, I have clarified this, and expanded the information on ADHD and inflammation.

I think, but cannot demonstrate, that the allergic tension fatigue syndrome did not crystalize because Speer was indeed describing patients with ADHD, but he could not know because: 1) at the fifties, the concept of ADHD was incipient; and 2) he was an allergologist, not a psychiatrist! I have commented this with the Head of Department of Allergology at my Hospital, Dr. Alfredo Iglesias-Cadarso, and he fully agrees with my perception.

3. The introduction would benefit from a more thorough review of prior research and original studies by Speer instead of relying primarily on the authors' interpretations.

RE: The reviewer is right. Accordingly, I have followed his/her recommendation of making a more thorough review of previous research, and minimizing my interpretations (that are later on discussed in the paper). I hope that now the introduction is more balanced and “fair”.  

4. The paper neglects to provide a comprehensive background on the role of histamine in the body and brain, an essential prerequisite to understanding the later arguments.

RE: Here, I think that the reviewer refers to the lack of information of histamine in the introduction, as I have extensively explained the role of histamine in the body and brain in point 3 (more than 4 pages on this, preparing readers for understanding my later arguments). If I’m right, I did not include info on histamine, as it was part of the hypotheses. However, I agree that it is important to include information on histamine in the introduction, and accordingly, have incorporated now, as suggested.

5. A discussion about the lack of conclusive proof regarding SATFS or the cholinergic imbalance is absent, which could lead to reader confusion

RE: Again, we agree. Accordingly, we have included information regarding this issue in the introduction.

6. The paper's complexity, in terms of medical jargon and advanced concepts, might impede comprehension for the general reader. A simpler presentation is recommended.

RE: The reviewer is right. I have been working more than a year on this paper trying to get related and comprehensible all these topics, and I agree that it is not an easy paper to follow. Accordingly, I agree with his/her suggestion. I think that all the points raised by the reviewer helped me to offer a simpler, easier to follow version of the paper. I tried to simplify the text and wiped out some jargon whenever possible. I hope that the current version of the paper is easier to follow. 

7. A clear, layman's definition of the immune response and inflammation would aid reader understanding, given their critical role in the discourse.

RE: Again, the reviewer is right. Accordingly, we have included this now (point 4.2.1).

8. A clear problem statement or research gap needs to be articulated earlier in the introduction.

RE: Again, the reviewer is right. I have amended this, and included this research gap need in the introduction.

9. The connection between histamine, immune response, and inflammation is not clearly established. Clarification or simplification would enhance reader comprehension.

RE: Again, the reviewer is right. Accordingly, we have changed the title and structure of point 4.2.2, and included a paragraph relating histamine, immune response and inflammation.

10. The correlation between immune response, inflammation, and ADHD, although apparent as the study focus, is not stated until later in the introduction, which disrupts the structure.

RE: Again, the reviewer is right. Accordingly, we have changed the structure of point 4.2.2., and included the relationship between immune response, inflammation, and ADHD before.

11. The introduction is laden with intricate biochemical relationship details, which might overwhelm a general audience. Relegating these to the methods or results section might help.

RE: The reviewer is right. As stated in the point 16, we have included more info on histamine, and simplified the information regarding the other biochemicals.

12. Diseases like cancer, cardiovascular disease, and autoimmune disorders are mentioned but not expounded upon, leaving readers unsure of their relevance to the study.

RE: I partially agree. This is true that I did not include that much info on cancer and cardiovascular disease, but I have included some info regarding autoimmunity. In any case, I take advantage on this comment to expand on the relationship between either cancer and cardiovascular disease and ADHD.

13. The significance of Speer syndrome is relegated to the introduction's end. If crucial to the research, it should be highlighted earlier with a more clear definition of its role.

RE: The reviewer is right. As pointed out in point 4, we have made extensive changes in the Introduction. Now, we have included an in-depth review of Speer work.

14. The hypothesis is not explicitly stated in the introduction, which might leave readers adrift.

RE: Again, the reviewer is right. Accordingly, we have rewritten the introduction in order to clearly state our hypotheses.

15. The discussion includes an array of neurotransmitters and molecules, which might confuse readers. Greater focus on the role of histamine would be beneficial.

RE: Again, the reviewer is right. Now, we have simplified the information regarding other neurotransmitters and molecules, and expanded the role of histamine.

16. The conclusion does not adequately summarize the study's findings, leaving the reader without a clear takeaway.

RE: Indeed, the reviewer is right, and this is a very important point. Accordingly, I have made extensive changes in the conclusion.

17. The paper concludes with a strong statement on the dearth of studies exploring histamine or DAO, and ADHD, which seems contradictory to the prior discussion of histamine's potential role.

RE: The reviewer is right. This paragraph was not well-written, the area where no info is available is the liaison DAO-ADHD. I have corrected this.

18. The link between SATFS and ADHD is not fully explored in the discussion, which could confuse the readers.

RE: Again, the reviewer is right. Now, I have included information regarding the link between SATFS and ADHD in the conclusion.

19. The claim that ADHD is primarily caused by an immune response to allergens lacks supporting evidence, creating a credibility issue.

RE: I agree with the reviewer that there is not evidence to state that ADHD is primarily caused by an immune response to allergens (if at all, that they are related processes). Although I don´t think that I was suggesting this with my paper, it is possible that I was translating into the text the words by Marshall, a paper which is critical for mine, and who asked himself in 1989 “if these behaviors are caused directly by allergic reactions, what are the biochemical or brain mechanisms involved in the pathogenesis of ADHD with an etiology of allergy?”. I state this clearly in the text. Also, I put down any information suggesting that ADHD is a byproduct of an immune response to allergens.

20. The paper lacks a comparative analysis with existing literature, which is essential to put the findings in context.

RE: The reviewer is right. This critic is related to point 2. See my response to this point. We have addressed this.

21. The conclusion fails to reiterate the significance of the study, leaving readers uncertain of the implications.

RE: The reviewer is right. We have included some comments on the implications. See also our response to point 17.

22. The paper would benefit from a more compelling concluding statement, encapsulating the research and its potential implications.

RE: The reviewer is right. We have followed his/her recommendations. We hope that now our conclusion encapsulates the research and potential implications.

23. In light of these points, I recommend a thorough review and revision of the manuscript to present a more compelling, organized, and substantiated discussion. Thank you for considering these comments.

RE: Thank you very much for your revision. It was to tough to address all the raised issues, but you were right in them. Accordingly, your critics have helped me to order my ideas, and I think that the current version is not only better structured, but easier to follow.

Kind Regards,

RE: Thanks indeed.